# RC-FCL: Combating Asynchronous Concept Drift in Federated Continual Learning via Retrospective Calibration

**Hang Su** [1]  **Yijun Mo** [1]  **Zhiyu Zhang** [1]  **Yankai Jiang** [2]  **Bo Liu** [3]  **Yichen Li** [1 4]  **Imran Razzak** [4]

## Abstract

Federated Continual Learning (FCL) enables the continuous acquisition of knowledge from streaming tasks, but inherently struggles with the temporal dynamics of client data distributions. These dynamics naturally induce asynchronous concept drift, where distribution shifts occur independently across clients at unsynchronized times and with varying magnitudes. Such asynchrony generates conflicting updates that destabilize global convergence and exacerbate catastrophic forgetting. However, existing FCL research focuses on static or incremental settings, typically treating all incoming updates uniformly, which obscures concept drift under divergent distributions and fails to adapt to the evolution of learned concepts. To address these limitations, we propose **RC-FCL**, a retrospective calibration framework for FCL that can effectively distinguish asynchronous concept drift and adjust the learning strategy adaptively. Specifically, RC-FCL leverages a conditional generative model to synthesize class-conditional reference distributions of previously learned concepts for local drift detection. It calibrates local adaptation using a weighting mechanism driven by the local discriminator to prioritize informative samples, and executes a global aggregation strategy based on drift magnitude. Our experimental results demonstrate that RC-FCL achieves competitive performance against state-of-the-art methods.

---

[1]School of Computer Science and Technology, Huazhong University of Science and Technology, Wuhan, China [2]Shanghai Artificial Intelligence Laboratory, Shanghai, China [3]Faculty of Computer Science and Artificial Intelligence, Shenzhen University of Advanced Technology, Shenzhen, China [4]Department of Computational Biology, Mohamed bin Zayed University of Artificial Intelligence, Abu Dhabi, United Arab Emirates. Correspondence to: Yichen Li <ycli0204@hust.edu.cn>, Yijun Mo <moyj@hust.edu.cn>.

*Proceedings of the 43$^{rd}$ International Conference on Machine Learning*, Seoul, South Korea. PMLR 306, 2026.

## 1. Introduction

Federated Learning (FL) is a distributed machine learning paradigm that enables multiple edge clients to collaboratively construct a unified global model without sharing their raw local data (Li et al., 2022b; Wang et al., 2022). Driven by the increasing need for data privacy, FL has garnered significant attention in areas like healthcare (Antunes et al., 2022; Wu et al., 2026) and mobile devices (Lim et al., 2020; Li et al., 2026).

However, most FL approaches assume static data distributions that remain unchanged over time (Karimireddy et al., 2020; Li et al., 2024a). This static assumption rarely holds in real-world applications, where each client can continue to collect new data and include it in the federated training (Li et al., 2024c). This continuous influx of new data introduces the challenge of catastrophic forgetting (Ganin et al., 2016), where the acquisition of new knowledge causes the model to overwrite and forget previously learned knowledge, leading to performance degradation on previous tasks. To break this limitation, Federated Continual Learning (FCL) (Yang et al., 2024) has been proposed, synthesizing principles from FL and Continual Learning (CL) (Wang et al., 2024a) to enable collaborative learning on streaming data.

Nevertheless, mitigating catastrophic forgetting alone is insufficient for realistic FCL scenarios. Beyond the sequential arrival of tasks, the data distributions associated with previously learned classes may themselves evolve over time, violating the implicit assumption of distributional stability adopted by most FCL methods (Li et al., 2025a). This phenomenon, commonly referred to as concept drift (Bayram et al., 2022), alters the statistical properties of existing classes without expanding the label space, rendering historical representations progressively suboptimal even when forgetting is controlled, in contrast to the explicit task boundaries typically assumed in continual learning.

Concept drift has been widely studied in the continual learning literature (Gomez-Villa et al., 2024), and recent efforts have begun to explore drift-aware learning in federated settings. These works address different aspects of the problem, such as detecting distributional changes, adapting models to new data, or improving robustness under non-stationary en-

vironments (Li et al., 2025b). Nevertheless, when deployed in FCL systems, concept drift introduces a set of intertwined challenges that are not simultaneously addressed by existing approaches. In particular, concept drift exhibits an additional layer of complexity in FCL due to its asynchronous nature. Specifically, distribution shifts may occur at different time steps, affect only a subset of clients, and vary in magnitude across clients. We summarize three key challenges arising from asynchronous concept drift in FCL: (1) drift signals are sparse and heterogeneous, and the inherent data heterogeneity across clients further obscures these signals, making it difficult to distinguish concept drift from normal variations in streaming tasks; (2) due to privacy risks, it is difficult for clients to obtain a reliable global distribution or retain previous tasks for effective drift detection; (3) the continuous evolution of concepts amplifies catastrophic forgetting, making it more challenging to preserve previous knowledge while adapting to new concepts effectively.

While existing FCL methods partially address individual aspects of these challenges—for example, by preserving past knowledge or improving robustness to data heterogeneity—they typically treat incoming client updates uniformly and lack mechanisms to explicitly calibrate learning under asynchronous distributional shifts. As a result, global models may fail to adapt effectively to evolving concepts while maintaining stable performance on previously learned data.

To address these challenges, we propose a retrospective calibration framework called RC-FCL that integrates concept drift calibration and continual learning to learn the new task better while mitigating asynchronous concept drift. Specifically, RC-FCL leverages a shared conditional generative model to retrospectively synthesize class-conditional reference distributions of previously learned data. Upon the arrival of a new task, clients compare local representations with these references in feature space to identify drifted classes. Based on the detected drift signals, RC-FCL calibrates local training through adaptive sample weighting and performs drift-aware aggregation at the server, assigning higher importance to clients experiencing more significant distributional shifts. This joint mechanism enables RC-FCL to effectively distinguish and adapt to concept drift while mitigating catastrophic forgetting and preserving privacy without storing historical data. Our main contributions can be summarized as follows:

- We systematically study the challenge of asynchronous concept drift in FCL scenario and analyze its impact on local optimization and global aggregation.

- We propose a unified RC-FCL framework for FCL that seamlessly integrates drift detection and adaptation with continual learning, enabling the model to preserve previous knowledge effectively.

- We conduct extensive experiments on various datasets and under different FCL task scenarios. The results demonstrate that our proposed model outperforms current state-of-the-art methods.

## 2. Related Work

**Federated Continual Learning.** Federated continual learning aims to integrate the privacy-preserving advantages of federated learning with the incremental knowledge acquisition capability of continuous learning. Early research (Yoon et al., 2021) decomposes model weights into global federated and sparse task-specific parameters, enabling clients to selectively integrate task-specific knowledge via weighted combinations. (Hu et al., 2024) demonstrates that layer-wise model recombination offers a more flexible mechanism for integrating heterogeneous client knowledge. (Zhang et al., 2023; Qi et al., 2023; Tran et al., 2024) employ replay-based methods, mixing old data when training on new tasks to mitigate catastrophic forgetting. (Rong et al., 2025) propose CAN, which leverages clients as navigators to guide generative replay, improving the quality of replayed samples and alleviating forgetting in federated continual learning. (Dong et al., 2022; Wang et al., 2023) introduce distillation methods to enhance knowledge fusion and transfer between tasks. (Babakniya et al., 2023; Wuerkaixi et al., 2024) leverage generative models for reconstructing past distributions. The former prioritizes privacy preservation by employing data free methods to train the model on the server after each task, eliminating the need for client data access. The latter combats client statistical heterogeneity and data noise through selective knowledge discarding. (Wang et al., 2024b) enhances repetitive task learning by decomposing models into labeled sub-models for client optimization, then precisely locating repetitive tasks through group aggregation to enable selective training. FedTA (Yu et al., 2025) addresses spatial-temporal data heterogeneity in FCL by anchoring tail distributions, enhancing robustness under heterogeneous and evolving client data streams.

**Concept Drift Detection**. Concept drift has been extensively studied in machine learning. (Gao et al., 2022) proposes a semi-supervised emerging class detection framework that identifies similar instances within local regions of the feature space via a mutual graph clustering mechanism, aiming to detect unknown classes in non-stationary data streams. (Li et al., 2022a) leverages the predictability of environmental evolution to model future concept drift trends in streaming data, anticipating distribution shifts and generating synthetic samples aligned with future distributions for model training. Other studies detect specialized concept drift, (Gower-Winter et al., 2025) proposes Performative Drift, which refers to the phenomenon where predictions made by deployed models influence the future distributions

they aim to predict. While these methods primarily target centralized machine learning, some research (Zhou et al., 2024; Panchal et al., 2023; Zhang et al., 2024) has shifted focus towards addressing concept drift in FL. However, these studies typically assume that all clients share identical conditional data distributions. In more realistic scenarios, different clients may undergo different concept drifts.

# 3. Methodology

## 3.1. Problem Formulation

We consider a federated continual learning system with a server and a set of $K$ clients. A model learns from a sequence of streaming tasks $D = \{D^1, D^2, \ldots, D^t\}$ where $D^t$ denotes the task at the step $t$. Each task $D^t$ can span multiple communication rounds to collaboratively optimize the global model. Each client $k$ maintains a local dataset $D_k^t = \{(x_i, y_i)\}$, where $x_i$ denotes a sample and $y_i \in Y^t$ is its corresponding label. Here, $Y^t$ represents the label space observed up to the $t$-th task. Within $D_k^t$, each label $y_i$ may belong to the set of previously encountered labels $\{Y^1, Y^2, \ldots, Y^{t-1}\}$ or correspond to a new label that emerges at the current step $t$. We denote the classification model by $w$ and the global model at step $t$ by $w^t$, while the local model at client $k$ is denoted by $w_k^t$. The objective is to optimize the global model $w^t$ such that it generalizes well to all tasks $D$, which reflects the overall distribution of data across all learned tasks. The goal can be formulated as:

$$\arg\min_{w^t} \sum_{j=1}^{t} \sum_{k=1}^{K} \frac{1}{|D_k^j|} \mathbb{E}_{x \sim P(x|y) \in D^j} \mathcal{L}(f(x; w_k^j), y),$$

$$\text{where} \quad w^t = \sum_{k=1}^{K} \alpha_k^t w_k^t. \tag{1}$$

where $\mathcal{L}$ denotes the cross-entropy loss, and $\alpha_k^t$ is the aggregation weight assigned to client $k$ at step $t$.

Next, we introduce the asynchronous concept drift challenge in FCL. Assuming that the concept drift occurs at $t$, if the joint probability distribution changes from $P_k^{t-1}(x, y)$ to $P_l^t(x, y)$ on different clients, i.e., $P_k^{t-1}(x, y) \neq P_l^t(x, y)$. We refer to the distribution before the drift as the pre-drift distribution, and the corresponding distribution after the drift as the post-drift distribution. To model asynchronous concept drift, we note that at step $t$ different clients may experience distributional shifts on different previously learned classes. As a result, the effective local objective varies across clients. We define the local training loss at client $k$ as

$$\mathcal{L}_k^t = \mathcal{L}_{\text{task}}^t + \mathcal{L}_{\text{replay}}^t + \sum_{y \in \mathcal{Y}^{t-1}} \gamma_k^t(y) \, \mathcal{L}_{\text{drift}}^t(y). \tag{2}$$

where $\mathcal{Y}^{t-1}$ denotes the set of previously observed classes. The coefficient $\gamma_k^t(y) \in [0, 1]$ indicates whether class $y$

at client $k$ undergoes concept drift at step $t$. Under asynchronous drift, $\gamma_k^t(y)$ varies across clients and classes, leading to heterogeneous local objectives across the federation. The mechanisms introduced in the subsequent sections are designed to implicitly capture and exploit this class-level variation for drift-aware learning in federated continual settings.

## 3.2. RC-FCL: Retrospective Calibration for FCL

The key idea of RC-FCL is to detect asynchronous concept drift and adaptively learn from the drifting samples. We describe our method through two modules: (1) asynchronous drift detection; and (2) personalized distribution-aware drift adaptation. These components provide a concrete realization of the conditional drift formulation in Eq. (2), enabling drift-aware learning under asynchronous concept drift. Specifically, each new task arrives with the server synthesizing data from previous tasks by an aggregated Conditional Generative Adversarial Network (CGAN) and broadcasting the global classification model and synthesized data to selected clients. The CGAN is trained in the FedAvg manner while aggregating the generator and keeping the discriminator locally before receiving the new task. Then, each client treats the synthesized data as reference distributions. Using Sinkhorn distance in the feature space, clients detect drifted classes by comparing current data with the synthetic reference. For identified drifted samples, the local discriminator assesses their novelty, and adaptive sample weights are assigned accordingly, where samples that deviate more from the pre-drift distribution receive higher weights. The local model is then updated using a weighted loss over both new and synthesized data. After training, clients send back updated models and drift statistics. Finally, the server performs a drift-aware aggregation, giving more weight to clients that experienced greater distributional changes, thereby ensuring global adaptation to asynchronous concept drift across clients. We illustrate the framework in Fig.1.

**Retrospective Drift Detection.** When the new task arrives, identifying which classes have experienced concept drift is crucial. This task is particularly challenging in FCL due to several fundamental limitations: (i) each client has only a partial view of the global data distribution; (ii) they are constrained by strict privacy policy and limited storage capacity, which makes it infeasible to store historical data from all tasks; (iii) the drift may occur at different steps across different clients, making it asynchronous in both temporal and spatial dimensions.

To overcome these challenges, each client requires a reliable reference that captures pre-drift concept distribution for comparison with current local data. We propose leveraging the shared global generative model to retrospectively synthesize reference distributions of previously learned concepts,

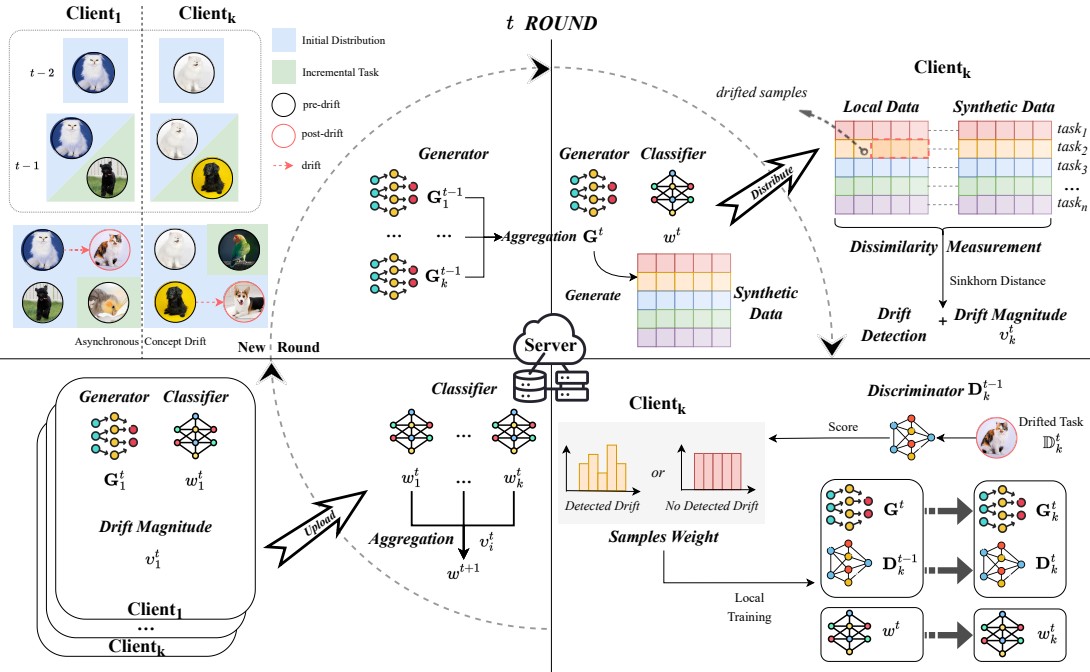

*Figure 1.* Overview of the RC-FCL framework. Upon the arrival of a new task, the server first synthesizes class-conditional data using an aggregated generator and distributes it along with the global model to selected clients. Each client detects concept drift by comparing local and synthetic feature distributions and assigns adaptive weights to drifted samples using a local discriminator. Finally, the server performs drift-aware aggregation based on the reported drift magnitudes to update the global model.

avoiding the need to store historical data. A substantial distributional divergence between local samples and synthetic samples generated to represent previously learned concepts serves as a strong indicator of concept drift. However, traditional generative models struggle to control the generation space, making it difficult to compare against new task data for concept drift detection. Inspired by (Qi et al., 2023), CGAN leverages label information to guide the generation process, enabling the synthesis of data from specific previous task distributions. This allows for a more targeted and meaningful comparison between the current and previous data distributions, facilitating reliable drift detection here. Before the arrival of the $(t+1)$-th new task, each client will additionally maintain a shared conditional generator $\mathbf{G}_k^t$ and a local discriminator $\mathbf{D}_k^t$, which are then optimized through an adversarial training objective, defined as:

$$\min_{\mathbf{G}_k^t} \max_{\mathbf{D}_k^t} \mathcal{L}_{\text{CGAN}} = \mathbb{E}_{(x,y)\sim D_k^t}\big[\log f(\mathbf{D}_k^t, x)\big] \quad (3)$$

$$+ E_{z\sim p_z(z), y\sim p_y(y)}\big[\log(1 - f(\mathbf{D}_k^t, \mathbf{G}_k^t(z,y)))\big]. \quad (4)$$

Then, the server will aggregate the shared conditional generator $\mathbf{G}_k^t$ like FedAvg to get $\mathbf{G}^t$. When arriving at the $(t+1)$-th new task, the server will synthesize previous data $\tilde{D}_k^t$ with $\mathbf{G}^t$ and broadcast to client $k$. For each label $y$

present in the local data $D_k^{t+1}$, the client checks if concept drift has occurred for $y$. Specifically, we use intermediate layers of the client model $w_k^t$ to extract feature representations from both the current local data $D_k^{t+1}$ and synthetic samples $\tilde{D}_k^t$. This process produces two sets of feature vectors corresponding to $D_k^{t+1}$ and $\tilde{D}_k^t$, which are treated as empirical distributions $P_{D_k^{t+1}}$ and $P_{\tilde{D}_k^t}$ in the feature space. We compute the distance between $P_{D_k^{t+1}}$ and $P_{\tilde{D}_k^t}$ in the representation space, and define a drift score that measures the magnitude of distributional shift:

$$\Delta_k^{t+1}(y) = \max\left(0, S_\epsilon(P_{D_k^{t+1}}, P_{\tilde{D}_k^t}) - \tau\right). \quad (5)$$

where $\Delta_k^{t+1}(y)$ quantifies the margin by distance exceeds a reference threshold $\tau$. Rather than enforcing a hard binary decision, the threshold $\tau$ serves as a baseline for characterizing drift strength. RC-FCL tolerates moderate inaccuracies in drift detection, as their effects are reflected through gradual adjustments in calibration strength. This behavior stems from the use of soft drift scores and prevalence-aware aggregation, which attenuate the impact of noisy drift signals. A positive drift score indicates a potential distributional shift, while larger values correspond to stronger drift. We consider samples associated with classes satisfying $\Delta_k^{t+1}(y) > 0$ as candidates for drift-aware calibration, forming the drifted subset $\mathbb{D}_k^{t+1}$.

We adopt the Sinkhorn distance for drift detection because it measures the minimal cost required to transform one feature distribution into another. This formulation aligns naturally with the nature of concept drift, which manifests as changes in class-conditional data distributions rather than isolated deviations of individual samples. By comparing distributions at the class level, the Sinkhorn distance captures structured shifts between pre-drift and post-drift representations. Moreover, it can be estimated directly from feature representations without introducing auxiliary detector networks, which fits naturally into the federated learning setting. The Sinkhorn distance is defined as follows:

$$S_\epsilon(M, N) = \min_{T \in \Pi(M,N)} \sum_{i,j} T_{ij} C_{ij} - \epsilon \sum_{i,j} T_{ij} \log T_{ij}.$$

where $T$ represents the transport plan between $M$ and $N$. The $\sum_{i,j} T_{ij} C_{ij}$ is the total transport cost, where $C_{ij}$ is the squared Euclidean distance between the feature vectors of the $i$-th local sample and the $j$-th synthetic sample. The entropic regularization term controlled by $\epsilon$ ensures numerical stability and efficient computation under minibatch estimation. This approach is effective due to its principled and localized design. By leveraging a conditional generative model as a reference, it enables drift detection without storing historical data, thereby meeting the strict privacy and storage constraints of FCL. The generative model produces class-specific reference distributions based on provided labels, which serve as the basis for identifying heterogeneous concept drift across different classes. Moreover, the generated data can be used for data replay to avoid catastrophic forgetting, which has been verified in many existing methods.

**Personalized Distribution-aware Drift Adaptation.** Upon detecting concept drift, the system must adapt to the evolving data distribution. A naive retraining strategy that treats all incoming samples equally is suboptimal, as it overlooks the varying contributions of different samples in capturing the similarities and differences between pre-drift and post-drift distributions. To address this issue, we propose a personalized distribution-aware drift adaptation method that pays more attention to samples and clients that can contribute more to the federated training at both local and global levels. At the client level, weights are assigned to individual samples to guide the model toward distinguishing features that separate the pre-drift concept from the post-drift one. At the server level, a differentiated aggregation strategy is employed to give greater influence to clients that have encountered more significant distributional shifts.

The local adaptation process begins by calculating the sample weights $\alpha_i$ for each sample $(x_i, y_i)$ in the detected shift dataset $\mathbb{D}_k^t$; we leverage the discriminator $\mathbf{D}_k^t$ to estimate its importance weight in the local update. Since the discriminator is trained on the pre-drift distribution, a low output score indicates a significant deviation from the pre-drift concept. Rather than aiming to precisely quantify samples level drift, we use the discriminator output as a soft proxy to bias learning toward more informative samples. Accordingly, we assign higher weights to samples with lower discriminator scores, making the weighting inversely related to the discriminator output:

$$\alpha_i = \begin{cases} 1 - f(\mathbf{D}_k^t, x_i), & x_i \in \mathbb{D}_k^t. \\ 1, & x_i \in D_k^{t+1} / \mathbb{D}_k^t \cup \tilde{D}_k^t. \end{cases} \quad (6)$$

With these weights established, each client updates its local classification model $w_k^{t+1}$ by minimizing a weighted loss over both current local samples and replayed non-shifted samples generated by the conditional generator, as follows:

$$w_k^{t+1} - \eta \sum_{x_i \in \mathcal{R}_k^t} \alpha_i \nabla \mathcal{L}(f(w_k^{t+1}; x_i), y_i), \quad (7)$$

$$\text{where } \mathcal{R}_k^t = \{(x_i, y_i) \in D_k^{t+1} \cup \tilde{D}_k^t | y_i \notin \mathbb{D}_k^t\}.$$

where $\eta$ is the learning rate, $\mathcal{L}$ denotes the cross-entropy loss. After local training, each participating client uploads its updated classification parameters to the server. Under asynchronous concept drift, client updates no longer provide equally reliable estimates of the evolving global objective, as distributional shifts may vary in both magnitude and consistency across clients. To account for this heterogeneity, we perform a drift-aware weighted aggregation that modulates client contributions based on both the strength of observed drift and its prevalence across clients. Specifically, for each class $y$, we compute its drift prevalence across participating clients as:

$$\rho^{t+1}(y) = \frac{1}{K} \sum_{k=1}^{K} \mathbb{I}\left[\Delta_k^{t+1}(y) > 0\right]. \quad (8)$$

This quantity reflects the degree of cross-client consistency of a class-level distributional shift. Using the prevalence, we define a client-level drift magnitude as:

$$m_k^{t+1} = \frac{\sum_{y \in \mathcal{Y}} \rho^{t+1}(y) \Delta_k^{t+1}(y)}{\sum_{y \in \mathcal{Y}} \rho^{t+1}(y) + \varepsilon_0}. \quad (9)$$

This formulation emphasizes drift signals that are both pronounced at a given client and consistently observed across the federation, while attenuating the amplification of rare drift, and explicitly strengthening drift signals that are consistently observed across multiple clients.

To preserve the contribution of clients with no detected drift, we assign them a default weight of 1. Based on these weights, the server updates both the global classification

model $w^{t+1}$ through weighted aggregation:

$$w^{t+1} = \sum_{k=1}^{K} \frac{v_k^{t+1}}{\sum_{j=1}^{K} v_j^{t+1}} \, w_k^{t+1}. \qquad (10)$$

$$\text{where } v_k^{t+1} = 1 + m_k^{t+1}.$$

The effectiveness of RC-FCL stems from the coordinated interaction between local adaptation and global aggregation. At the client level, the discriminator provides a soft signal to prioritize informative post-drift samples during local updates, while the generator enables data-free drift detection and generative replay to mitigate forgetting. At the server level, aggregation weights are adjusted based on both the magnitude and prevalence of detected distributional shifts, preventing rare local variations from disproportionately influencing the global model. RC-FCL reflects a conservative bias toward stable global optimization under asynchronous drift, while allowing emerging concepts to gradually gain influence as they become more prevalent. Together, these mechanisms allow RC-FCL to adapt to evolving concepts in a stable yet responsive manner under asynchronous drift. The algorithm flow of RC-FCL is in Appendix B .

# 4. Experiments

## 4.1. Setup

**Dataset:** We conduct experiments on four image datasets with heterogeneous dataset partition: CIFAR10, CIFAR100 (Krizhevsky et al., 2009) Digit10, and Tiny-ImageNet (Le & Yang, 2015). To introduce concept drift, our goal is not merely to degrade image quality (e.g., by making images blurry), but to construct a consistent post-drift distribution. To this end, we employ a set of controlled transformations, including Gaussian noise, global rotation, local rotation, and random erasing. These transformations are carefully designed to create a post-drift distribution that preserves partial similarities with the original distribution while simultaneously introducing distinct characteristics introduced by the drift. Details of the datasets and data processing can be found in Appendix A.

**Metrics:** We evaluate the performance of methods in the federated continual learning setting using two metrics: Average Accuracy (Acc) and Average Forgetting Score (FS). The former assesses the overall classification performance by computing the average accuracy over the entire test set after completing the entire task stream, reflecting the model's ability to learn new tasks while preserving prior knowledge. The latter quantifies the extent of forgetting by comparing the maximum accuracy achieved during training for each task with its corresponding final accuracy after completing all tasks. In concept drift, evaluations are conducted on the post-drift data distribution, ensuring that the classifier

adapts to the most recent distributional characteristics.

**Baseline:** We compare RC-FCL with a variety of baseline methods, including those specifically designed for FCL and CL methods adapted to the federated setting. These baselines fall into three general categories. The first group leverages **replay mechanisms** to mitigate forgetting, represented by methods such as FedCIL (Qi et al., 2023), Re-Fed (Li et al., 2024b), CAN (Rong et al., 2025) and FedCBDR (Qi et al., 2026). A second line of work adopts **regularization strategies** to preserve previously learned knowledge, including FedWeIT (Yoon et al., 2021), FedSSI (Li et al., 2025c), and FedEWC (Kirkpatrick et al., 2017). The third category incorporates knowledge **distillation techniques** to transfer information between tasks or clients, as seen in FedLwF (Li & Hoiem, 2017), Target (Zhang et al., 2023), and GLFC (Dong et al., 2022). In addition, we include FLASH (Panchal et al., 2023) and FedTA (Yu et al., 2025) for handling distribution shifts.

**Configurations:** Unless otherwise mentioned, we set the number of local training epochs $E = 20$ and communication rounds $T = 100$ for each task, which ensures the convergence of previous tasks before the arrival of a new task. The client number $K = 20$ with an active ratio $r = 0.4$. For local training, the batch size is 32 and the learning rate is 0.001 for training the classifier, generator, and discriminator. We use the Dirichlet distribution $\text{Dir}(\alpha)$ on labels to simulate the data heterogeneity, where a smaller $\alpha$ indicates higher data heterogeneity. For the classifier in all methods, we employ ResNet18 (He et al., 2016) as the basic backbone. The generator consists of a fully connected layer followed by three convolutional blocks with BatchNorm and LeakyReLU. The discriminator adopts a similar structure but is designed to output probabilities instead of images. Unless otherwise specified, we set Dirichlet distribution $\text{Dir}(\alpha)$ to 0.5, the threshold $\tau = 0.4$, and the entropy regularization coefficient $\epsilon = 0.05$. We adopt a 10-task setting for CIFAR100 and Tiny-ImageNet and a 5-task setting for CIFAR10 and Digit10. At the arrival of each new task, one previously learned task is randomly selected to undergo a specific type of concept drift. All experiments are conducted on GeForce RTX 4060 GPU.

## 4.2. Performance Overview

**Test Accuracy and Forgetting Score**. Table 1 presents the main experimental results, comparing RC-FCL with twelve baselines on two datasets. Evaluations are conducted under three different data partitioning schemes to simulate varying degrees of data heterogeneity. We report the global model's average accuracy and forgetting score after all clients have completed training on the entire task stream. The results clearly demonstrate the superiority of our method. Across all levels of heterogeneity, RC-FCL consistently outper-

*Table 1.* Performance comparison of various methods on two datasets. The best results are **bold**.

| Dataset | CIFAR100 | | | | | | Tiny-ImageNet | | | | | |
|---|---|---|---|---|---|---|---|---|---|---|---|---|
| Partition | IID | | $\alpha = 0.5$ | | $\alpha = 0.1$ | | IID | | $\alpha = 0.5$ | | $\alpha = 0.1$ | |
| Metric | Acc(↑) | FS(↓) | Acc(↑) | FS(↓) | Acc(↑) | FS(↓) | Acc(↑) | FS(↓) | Acc(↑) | FS(↓) | Acc(↑) | FS(↓) |
| FedCIL | 24.01 | 37.76 | 21.67 | 39.28 | 20.12 | 40.01 | 22.66 | 39.53 | 19.97 | 41.11 | 18.63 | 43.76 |
| ReFed | 29.16 | 33.52 | 27.51 | 38.83 | 26.28 | 39.54 | 24.67 | 38.02 | 20.49 | 40.24 | 19.86 | **38.45** |
| CAN | 28.72 | 39.67 | 25.67 | 38.82 | 24.93 | 40.78 | 24.29 | **37.91** | 22.18 | 39.96 | 21.07 | 40.46 |
| FedCBDR | 26.97 | 38.76 | 23.72 | 39.88 | 20.54 | 43.25 | 21.68 | 41.05 | 19.76 | 43.37 | 19.53 | 42.31 |
| FedEWC | 13.32 | 68.27 | 9.31 | 73.43 | 9.52 | 71.82 | 11.20 | 62.34 | 8.7 | 63.06 | 7.94 | 64.32 |
| FedWeIT | 14.50 | 63.81 | 11.96 | 67.45 | 9.33 | 68.50 | 9.85 | 62.11 | 9.02 | 65.73 | 8.42 | 65.36 |
| FedSSI | 21.17 | 40.28 | 20.34 | 44.19 | 18.40 | 46.63 | 18.81 | 42.39 | 17.08 | 45.91 | 15.58 | 46.54 |
| FLASH | 31.29 | 32.17 | 28.32 | 38.64 | 26.17 | 39.52 | 24.32 | 38.73 | 20.78 | 43.72 | 18.72 | 44.56 |
| FedTA | 32.14 | 34.92 | 28.68 | 37.10 | 24.54 | 39.03 | 25.06 | 40.12 | 21.71 | 41.25 | 20.92 | 43.19 |
| FedLwF | 20.99 | 43.59 | 18.71 | 42.44 | 14.52 | 50.32 | 16.15 | 46.59 | 14.23 | 51.63 | 12.74 | 55.28 |
| GLFC | 23.98 | 35.27 | 20.88 | 39.10 | 20.41 | 44.32 | 22.35 | 40.17 | 18.69 | 43.56 | 16.55 | 43.03 |
| Target | 28.72 | 39.06 | 25.49 | 43.53 | 24.11 | 43.02 | 24.14 | 41.20 | 21.26 | 42.38 | 20.71 | 45.66 |
| **RC-FCL** | **35.23** | **29.65** | **32.01** | **35.24** | **29.77** | **37.59** | **26.69** | 39.42 | **24.23** | **38.49** | **22.62** | 40.11 |

*Table 2.* Computational and communication costs.

| Method | Memory (MB) | Time (s/round) | Upload (MB/round) | Download (MB/round) |
|---|---|---|---|---|
| ReFed | 247.56 | 325 | 44.72 | 44.72 |
| GLFC | 312.91 | 367 | 44.72 | 44.72 |
| CAN | 301.36 | 405 | 56.02 | 56.02 |
| RC-FCL | 295.22 | 389 | 56.11 | 56.11 |

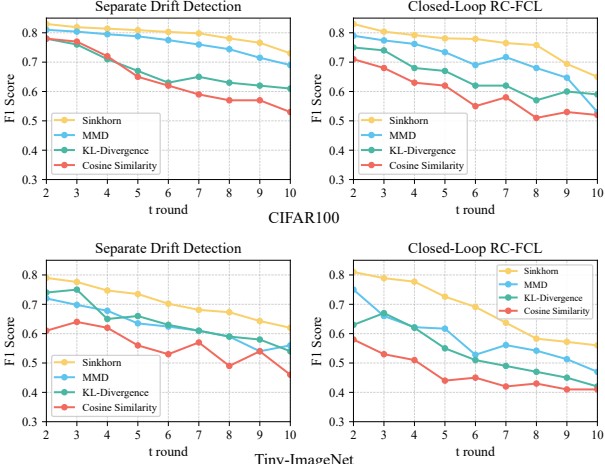

*Figure 2.* Comparison of drift detection performance (F1 Score) using different distance metrics on CIFAR100 and Tiny-ImageNet under Separate Drift Detection and Closed-Loop RC-FCL settings.

forms all baselines, achieving the highest accuracy and relatively lower forgetting. Furthermore, in our experiments, replay-based methods generally outperform regularization-based ones due to their superior model plasticity. While regularization approaches aim to preserve previously acquired knowledge by strictly constraining important parameters, this rigid preservation hinders the adaptability to concept drift. In contrast, replay methods jointly train on new and previous data, enabling more flexible model updates that facilitate learning new concepts while maintaining stable performance on earlier tasks.

**Asynchronous drift detection.** To evaluate whether the choice of distance metric genuinely improves drift detection, as opposed to merely benefiting from downstream adaptation, we designed two complementary experimental configurations. In the first configuration, we isolate the drift detection mechanism by using a shared reference generator and replay pipeline, while varying the drift distance metric to compute class-level drift scores. This setup allows us to assess the discriminative power of the metric independent of the impact of model updates. In the second configuration, we run the complete RC-FCL pipeline, where drift detection directly influences sample weighting and aggregation, and the reference generator is updated across tasks. This configuration represents a realistic closed-loop scenario typical in federated continual learning. Fig. 2 reports the F1 scores using drift labels induced by data transformations. Overall, Sinkhorn remains the most robust under both configurations, with its advantage becoming particularly clear in the end-to-end setting, where detection quality directly governs model

*Table 3.* Evaluation of various methods in terms of the communication rounds to reach the best test accuracy. We report the sum of communication rounds required to achieve the best performance on each task and evaluate with the trade-offs between communication rounds and accuracy. We denote "Δ" as the difference between the accuracy improvement percentage and the round increase percentage of RC-FCL and other baselines.

| Dataset | CIFAR10 | | CIFAR100 | | Tiny-ImageNet | | Digit10 | |
|---|---|---|---|---|---|---|---|---|
| Metric | Round | Δ | Round | Δ | Round | Δ | Round | Δ |
| FedCIL | $315_{\pm1.94}$ | 8.98%↑ | $711_{\pm1.71}$ | 39.84%↑ | $824_{\pm2.25}$ | 21.09%↑ | $255_{\pm1.18}$ | 4.70%↑ |
| ReFed | $344_{\pm2.76}$ | 10.82%↑ | $781_{\pm2.65}$ | 18.15%↑ | $846_{\pm2.71}$ | 20.62%↑ | $271_{\pm1.47}$ | 8.23%↑ |
| FedSSI | $331_{\pm1.57}$ | 10.42%↑ | $771_{\pm1.23}$ | 57.89%↑ | $831_{\pm2.34}$ | 42.46%↑ | $269_{\pm2.33}$ | 6.52%↑ |
| GLFC | $347_{\pm1.68}$ | 15.08%↑ | $737_{\pm2.47}$ | 49.23%↑ | $816_{\pm2.99}$ | 28.42%↑ | $275_{\pm1.69}$ | 9.87%↑ |
| Target | $329_{\pm2.36}$ | 7.30%↑ | $750_{\pm2.13}$ | 23.31%↑ | $851_{\pm1.69}$ | 16.91%↑ | $264_{\pm1.94}$ | 3.88%↑ |
| **RC-FCL** | $323_{\pm1.97}$ | / | $767_{\pm1.56}$ | / | $826_{\pm2.18}$ | / | $257_{\pm1.34}$ | / |

adaptation.

**Resource Consumption**. Table 3 compares the communication efficiency of different methods by examining the trade-off between the number of communication rounds and the resulting accuracy. We conducted experiments on both short task streams and long task streams to comprehensively evaluate method adaptability. While our method may introduce slightly more communication rounds compared to others, we argue that this additional cost is justified. With a limited increase in communication, our approach achieves significantly higher accuracy, making it an overall efficient solution. In particular, for the more challenging long task stream, our overall performance shows a substantial improvement compared with other methods. On relatively simple tasks such as Digit10, our method shows less advantage, likely because all approaches can already achieve high accuracy—even in the presence of concept drift—leaving limited room for further improvement and diminishing the relative benefits of advanced drift-handling mechanisms. Table 2 further reports peak memory, runtime, and per-round communication costs. RC-FCL introduces additional communication overhead due to its generative components while maintaining reasonable memory usage and runtime, suggesting that its improved performance is obtained with acceptable resource consumption.

**Sensitivity to the Number of Drifted Clients.** We investigate how varying the number of drifted clients (from 1 to 8) affects model performance. This experiment simulates a common real-world scenario, where new data patterns typically emerge within a small subset of users before gradually spreading across the entire population. As illustrated in Fig. 3, all methods benefit from more drifted clients due to stronger adaptation signals. This is because a larger number of drifted clients can provide the central server with a stronger and more explicit adaptation signal, enabling the global model to more effectively learn and adjust to the

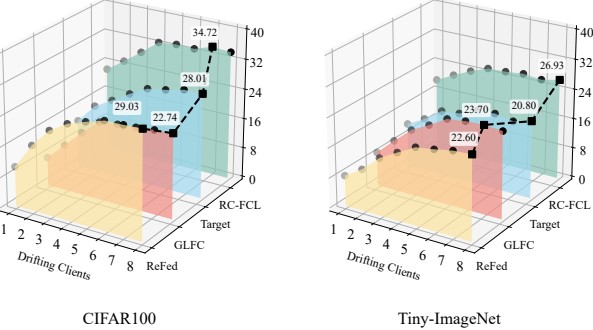

CIFAR100      Tiny-ImageNet

*Figure 3.* Performance comparison of various baselines under varying numbers of drifted clients.

new data distribution. Notably, RC-FCL maintains superior performance even with a few drifted clients, demonstrating low sensitivity to their proportion. The performance of other methods is significantly affected, as the valuable signals from a few drifted clients are overwhelmed by the updates from a large number of non-drifted clients, preventing the global model from perceiving this crucial distributional change. This robustness stems of RC-FCL from its ability to identify and prioritize drifted clients during aggregation, allowing effective global adaptation even when drift signals are sparse.

**Ablation Study.** Figure 4 investigates the importance of drift detection and drift adaptation components in our method across different datasets and experimental settings. The absence of either component leads to a noticeable drop in overall performance. In comparison, drift adaptation proves to be more critical, as it directly governs the model's ability to respond to distributional changes. Without effective adaptation, even accurate detection offers limited benefit, since the model cannot realign itself with the new concept.

Additional results in Appendix C examine RC-FCL under multiple concept drifts, hyperparameter sensitivity, and drift

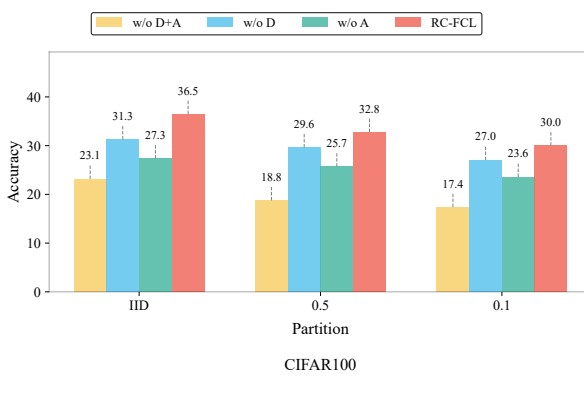

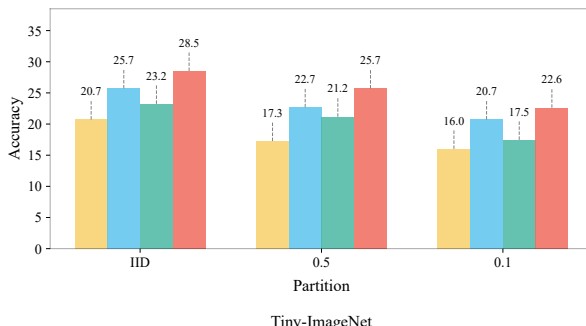

*Figure 4.* Ablation study exploring the importance of drift detection (D) and drift adaptation (A) across different settings

performance, complementing the main evaluation.

## 5. Conclusion

This paper presents RC-FCL, a novel retrospective calibration framework designed to address the challenges of asynchronous concept drift in federated continual learning. By leveraging conditional generative models and feature-space comparisons, RC-FCL effectively identifies concept drifts without requiring access to historical data. It incorporates adaptive weighting and drift-aware aggregation to ensure robust local adaptation and global model updates. Extensive experiments across multiple datasets and drift scenarios verify the effectiveness of RC-FCL.

## Acknowledgments

This work is supported by Hubei intelligent edge computing research institute, Hubei science and technology talent service project (2024DJC078); Interdisciplinary Research Program of HUST (5003210068); and the National Natural Science Foundation of China under grants 625B2073.

## Impact Statement

This paper presents work whose goal is to advance the field of Machine Learning. There are many potential societal consequences of our work, none which we feel must be specifically highlighted here.

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

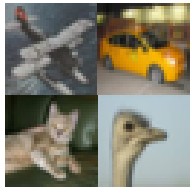 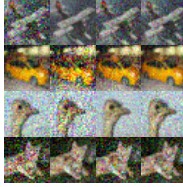 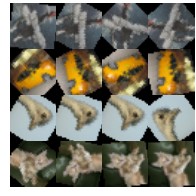 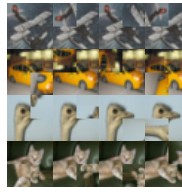 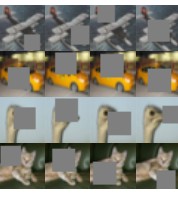

**(a) Original**     **(b) Gaussian Noise**     **(c) Global Rotation**     **(d) Local Rotation**     **(e) Random Erasing**

*Figure 5.* Visualization of simulated concept drifts applied to CIFAR10 samples. From left to right: (a) Original samples representing the pre-drift distribution; (b) Gaussian Noise simulating sensor degradation; (c) Global Rotation simulating viewpoint changes; (d) Local Rotation representing complex structural shifts; and (e) Random Erasing simulating object occlusion. These transformations generate the asynchronous concept drifts used in our experiments.

## A. Experimental Details

### A.1. Datasets

**CIFAR10**: A dataset with 10 object classes, including various common objects, animals, and vehicles. It consists of 50,000 training images and 10,000 test images.
**CIFAR100**: Similar to CIFAR10, but with 100 fine-grained object classes. It has 50,000 training images and 10,000 test images.
**Tiny-ImageNet**: A subset of the ImageNet dataset with 200 object classes. It contains 100,000 training images, 10,000 validation images, and 10,000 test images.
**Digit10**: A dataset contains 10 digit classes across diverse styles, such as handwritten digits and street numbers.

### A.2. Concept drift setups

Modeling concept drift in real-world federated environments is inherently challenging, as true data-generating processes are complex, non-stationary, and often unobservable. Rather than attempting to exhaustively reproduce real-world drift patterns, we adopt a controlled simulation protocol that isolates representative sources of distributional change while preserving class semantics. This design enables systematic evaluation, reproducibility, and principled analysis of drift-aware federated continual learning methods.

We simulate concept drift by applying a set of input-level transformations that modify the data distribution without changing the underlying class labels. Specifically, we consider four types of transformations:

1. Gaussian noise with variance ranging from 0.1 to 0.4, modeling sensor noise and environmental interference;

2. Global rotation with angles between $10°$ and $350°$, capturing changes in camera orientation or viewpoint;

3. Local rotation, where a randomly selected quarter-sized patch is rotated by $90°$, $180°$, or $270°$, representing localized structural variations;

4. Random erasing, where a contiguous region covering 10% to 30% of the image is replaced by a constant gray value, simulating partial occlusions.

These transformations are commonly used in prior work to study distribution shifts and robustness in visual recognition systems.

Importantly, our goal is not to claim that these controlled transformations fully capture the complexity of real-world concept drift. Instead, they serve as canonical and interpretable perturbations that reflect key characteristics of practical drift, such as gradual corruption, geometric variation, partial information loss, and heterogeneous impact across clients. By controlling the type and extent of distributional changes, we are able to disentangle the effect of drift detection and adaptation mechanisms from confounding factors and to reliably evaluate the robustness of federated continual learning under evolving data distributions.

---

**Algorithm 1** RC-FCL

---

**Input:** $T$: communication round; $K$: client number; $\eta$: learning rate; $\tau$: threshold; $\epsilon$: entropy regularization coefficient; $D$: streaming tasks;

**Output:** target classification model $w^{t+1}$.

1: **In Server**:
2: Initialize the parameter $w$;
3: **for** $t = 1$ **to** $T$ **do**
4:     Aggregate generator $\mathbf{G}_k^{t-1}$ to get $\mathbf{G}^t$ ;
5:     Generate synthesized data via $\mathbf{G}^t$;
6:     randomly select clients from $K$ total clients;
7:     **for** each selected client **in parallel do**
8:         sends $w^t$, $\mathbf{G}^t$ and synthesized data to the client;
9:         receives $w_k^t$, $\mathbf{G}_k^t$ and $\upsilon_k^t$ from the client;
10:     **end for**
11:     Aggregates classification model $w_k^t$ with weights $\upsilon_k^t$ to get $w^t$;
12: **end for**
13: **In each selected client** $C_k$:
14: Set local models $w_k^t = w^t$, $\mathbf{G}_k^t = \mathbf{G}^t$;
15: Detect drift by computing $S_\epsilon$ with (5);
16: Compute drift sample weights $\alpha_i$ using discriminator $\mathbf{D}_k$ with (6);
17: Compute mean drift magnitude $\upsilon_k^t$ with (9)(10);
18: Updates classification model $w_k^t$, $\mathbf{G}_k^t$, and $\mathbf{D}_k$ using weighted samples with (3)(4)(7);
19: Push the $w_k^t$, $\mathbf{G}_k^t$ and $\upsilon_k^t$ to the server;

---

Unless otherwise specified, drift detection experiments uniformly mix the four transformation types across clients and tasks. This mixed-drift protocol prevents methods from overfitting to a specific transformation pattern and better reflects the diversity of distribution shifts encountered in federated environments.

## B. Algorithm

The RC-FCL process is shown in algorithm 1.

## C. Additional Results

### C.1. Parameter Sensitivity Analysis.

**Parameter Sensitivity Analysis.** We conduct a deeper investigation into the impact of two key hyperparameters: the threshold $\tau$ and the entropy regularization coefficient $\epsilon$, both of which influence the accuracy of drift detection and subsequent adaptation. The threshold controls detection sensitivity—if set too low, it leads to excessive false positives by overreacting to minor statistical noise; if too high, it causes false negatives by missing real drifts. The regularization coefficient balances the precision and stability of distance computation. While smaller values may improve approximation accuracy, they can also introduce numerical instability and slow convergence. As shown in Fig.6, an appropriately small $\epsilon$ and a well-chosen $\tau$ result in reliable and stable drift detection, enabling effective adaptation to concept changes.

### C.2. Pre-drift and Post-drift

**Comparison on Concept Drift Distributions.** Fig.7 illustrates the model performance before and after concept drift across selected tasks from CIFAR10, CIFAR100, and Tiny-ImageNet. Specifically, we select four drifted tasks from CIFAR10 and eight from the other two datasets. As shown in the figure, RC-FCL demonstrates superior accuracy on the post-drift distribution while exhibiting slightly lower accuracy on the pre-drift data. The pre-drift curve is entirely enclosed within the post-drift curve, indicating a successful adaptation to the new distribution. In contrast, Re-Fed shows an interleaved pattern between pre-drift and post-drift accuracy, suggesting its inability to distinguish between the two distributions effectively. This implies that Re-Fed remains influenced by outdated knowledge, resulting in less consistent performance after drift.

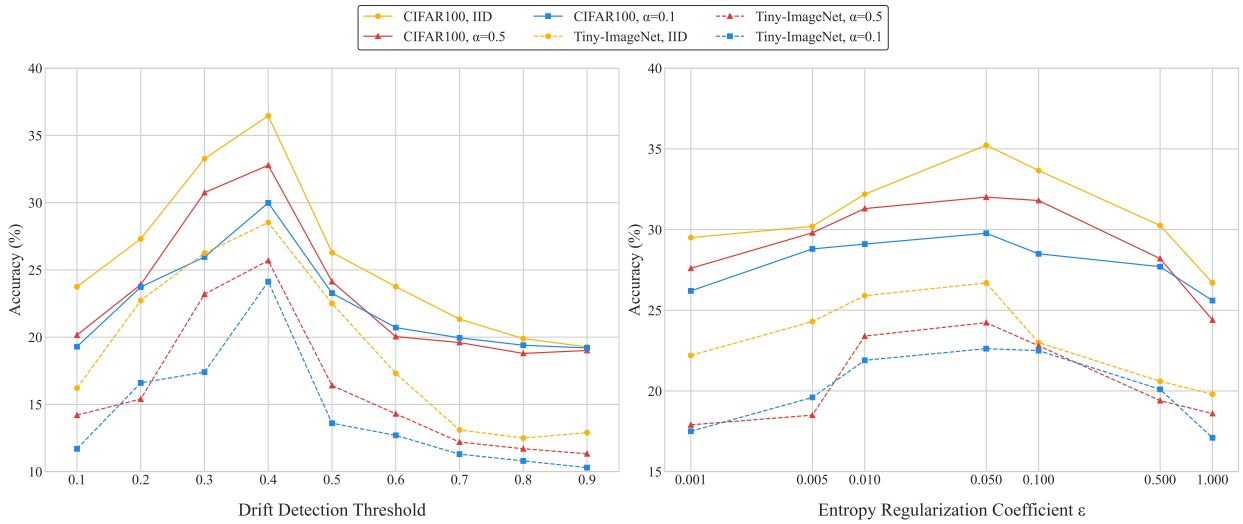

*Figure 6.* Sensitivity analysis of the threshold $\tau$ and entropy regularization coefficient $\epsilon$ in the drift detection.

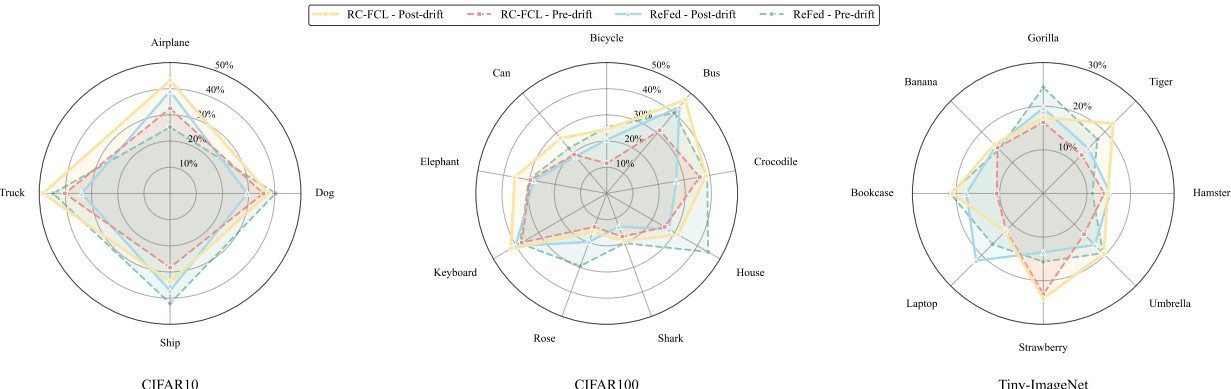

*Figure 7.* Performance of RC-FCL and Re-Fed on pre-drift and post-drift test data for drifted classes from three datasets.

*Table 4.* Comparison of model performance under multiple sequential concept drifts on the CIFAR10 dataset.

| Method | FedCIL | | ReFed | | FedLwF | | GLFC | | Target | | **RC-FCL** | |
|---|---|---|---|---|---|---|---|---|---|---|---|---|
| Metric | Acc($\uparrow$) | FS($\downarrow$) | Acc($\uparrow$) | FS($\downarrow$) | Acc($\uparrow$) | FS($\downarrow$) | Acc($\uparrow$) | FS($\downarrow$) | Acc($\uparrow$) | FS($\downarrow$) | Acc($\uparrow$) | FS($\downarrow$) |
| Drift-1 | 45.19 | 26.11 | 42.57 | 27.44 | 37.55 | 30.67 | 39.03 | 28.45 | 43.42 | 24.03 | 49.78 | 25.16 |
| Drift-2 | 31.26 | 31.39 | 35.92 | 31.99 | 29.01 | 32.93 | 35.66 | 30.23 | 40.83 | 29.22 | 47.95 | 29.28 |
| Drift-3 | 27.23 | 34.6 | 30.15 | 30.81 | 24.44 | 36.57 | 27.29 | 29.44 | 33.74 | 30.15 | 47.63 | 27.19 |

## C.3. Robustness under Multiple Concept Drifts.

**Robustness under Multiple Concept Drifts.** Table 4 presents results on CIFAR10 with three sequential concept drifts: random erasing, Gaussian noise, and global rotation. These three types of drifted data for the same task appear randomly in the local data at different times. Drift-3 indicates that the data distribution of a given task undergoes three shifts throughout the entire training process. RC-FCL consistently achieves high accuracy, demonstrating strong adaptability to evolving distributions. In contrast, baseline methods suffer performance degradation as drift accumulates, due to their inability to discard outdated knowledge. Without explicit drift-handling mechanisms, they retain obsolete concepts, leading to confusion and hindered adaptation. By contrast, RC-FCL detects drift and prioritizes new samples, ensuring effective learning in dynamic environments.

