# OpenReview forum: "RC-FCL: Combating Asynchronous Concept Drift in Federated Continual Learning via Retrospective Calibration"
_ICML.cc/2026/Conference — ICML 2026 regular_

### Official Review · Reviewer_yo2X · 2026-03-08

**Soundness:** 4
**Presentation:** 3
**Significance:** 4
**Originality:** 3
**Overall Recommendation:** 5
**Confidence:** 5

**Summary:**

This paper addresses the problem of Federated Continual Learning under the specific challenge of Asynchronous Concept Drift. To mitigate this issue, the authors introduce RC-FCL, a framework combining three components: (1) a  generative model for privacy preserving data replay; (2) a distance metric for detecting drift ; (3) a dual adaptation mechanism (local weighting and global aggregation). Extensive experiments are conducted to validate the effectiveness of the proposed method.

**Compliance With Llm Reviewing Policy:**

Affirmed.

**Final Justification:**

After carefully reviewing the authors’ rebuttal, I am satisfied that my main concerns have been adequately addressed:

1. The authors clarified the training schedule of the CGAN relative to the classifier, explaining the order of updates and how it integrates with local training.
2. The additional computational costs, including maintaining the CGAN and computing Sinkhorn distances, were quantified, alleviating my concerns about practical feasibility.
3. While minor writing clarity issues remain, the authors’ explanations regarding the transition from drift detection to adaptation are now clear and understandable.

Other reviewers raised concerns regarding novelty, the quality of synthetic references, simulation of concept drift, and baseline comparisons. The authors provided detailed responses, including:

- Explaining the asynchronous concept drift formulation and its practical significance.
- Justifying the use of CGANs for class-specific replay and drift detection, and discussing alternatives.
- Extending baseline comparisons with recent 2025 studies and clarifying experimental settings.
- Reporting overhead and performance metrics, confirming scalability and efficiency.

Considering these clarifications, I am convinced that the methodology, experimental design, and presentation sufficiently address all major concerns. Therefore, I increase my score to 5, reflecting that the paper now meets the expected standards for clarity, soundness, and significance.

**Key Questions For Authors:**

See the weaknesses.

**Limitations:**

yes

**Strengths And Weaknesses:**

Strengths：
- The paper presents a novel approach for asynchronous concept drift in FCL, integrating existing techniques and introducing personalized, distribution-aware adaptation to handle heterogeneous client dynamics.

- The paper is clearly and effectively presented, with informative figures and tables; Figure 1 provides a concise overview of the framework, and the mathematical notation is consistent and easy to follow.

- Extensive experiments are conducted across diverse datasets, varying heterogeneity, and different drift scenarios, accompanied by ablation analyses.

Weaknesses：
- Although the structure is clear, the writing could be more concise in places. For example, the explanation of the design rationale during the transition between drift detection and adaptation could be tightened.
- While the paper evaluates the method in terms of communication rounds, it would be much stronger if the authors discussed the additional computational costs of maintaining CGANs and computing Sinkhorn distances.
- Could the authors clarify the training schedule for the CGAN relative to the classifier? Specifically, are the GAN and the classifier trained concurrently, alternately, or sequentially during local updates.

---

> ### Author Rebuttal · Authors · 2026-03-30
>
> > **W1. Issues about writing conciseness and presentation**
>
> **R1:** Thank you for this helpful suggestion. We agree that this transition can be written more concisely. The intended rationale is that drift detection identifies which classes or samples deviate from the synthetic pre-drift reference, and these signals are then directly used to drive local sample weighting and server-side drift-aware aggregation.  We will tighten this part in the revision to make the detection-to-adaptation logic more direct and easier to follow.
>
> > **W2. Concern about computational costs**
>
> **R2:**  Thank you for raising this concern. We conducted an additional overhead experiment.
>
> |Method|Peak Memory (MB)|Time (s)/round|Upload/round (MB)|Download/round (MB)|
> |-|-:|-:|-:|-:|
> |ReFed|247.56|325|44.72|44.72|
> |GLFC|312.91|367|44.72|44.72|
> |CAN[1]|301.36|405|56.02|56.02|
> |RC-FCL|295.22|389|56.11|56.11|
>
> The generative module is a multipurpose choice, as it is used for replay and drift adaptation, while discriminator is reused. Compared with memory buffer methods, we do not require storing raw data. Although we introduce a slight additional cost, it remains a cost-effective design.
>
> > **W3. Concern about training schedule**
>
> **R3:** Thank you for the thoughtful concern. In RC-FCL, the CGAN and the classifier are not intended to be optimized fully concurrently on the same step from the start of local updates. Instead, their roles are partially sequential. Before the arrival of task $t+1$, each client has already trained its local CGAN on task $t$, and the server aggregates the generator to synthesize pre-drift reference data. When task $t+1$ arrives, drift detection is first performed by comparing current local data with these synthetic references in feature space, and the resulting drift signals are used to compute sample weights for adaptation. The classifier is then updated using the weighted loss over current and replayed samples. Afterward, the generator and discriminator are further updated during local training and the updated generator is uploaded to the server for the next step.
>
> Therefore, RC-FCL adopts a partially sequential design. The CGAN first provides the pre-drift reference for drift detection and calibration, and the classifier is then updated based on these signals. The CGAN is subsequently updated for future rounds/tasks, rather than co-evolving throughout the same detection step. We will clarify this in the revision.
>
> > **References**
>
> [1]CAN: Leveraging Clients As Navigators for Generative Replay in Federated Continual Learning. (ICML 2025)

---

> > ### Author Rebuttal · Reviewer_yo2X · 2026-04-02
> >
> > Thanks for the authors' detailed responses. In general, the authors' rebuttal has satisfactorily addressed my concerns, specifically regarding the computational cost and training schedule.
> >
> > I have also carefully reviewed the other reviewers' comments and the authors' corresponding responses. I believe the authors have satisfactorily addressed the issues.
> >
> > Overall, I am satisfied with the rebuttal and recommend acceptance with an increased score of 5.

---

> > > ### Author Response · Authors · 2026-04-04
> > >
> > > Thank you for your recognition of our work and your valuable feedback. We will carefully refine the manuscript based on these discussions in the updated version.

---

### Official Review · Reviewer_uYMd · 2026-03-09

**Soundness:** 3
**Presentation:** 3
**Significance:** 4
**Originality:** 4
**Overall Recommendation:** 5
**Confidence:** 5

**Summary:**

The paper proposes RC-FCL to address asynchronous concept drift in Federated Continual Learning（FCL). Specially, RC-FCL utilizes a shared conditional GAN to generate reference distributions of past concepts without storing raw data. The method detects shifts using Sinkhorn distance in the feature space and prioritizes specific samples via a local discriminator. Finally, the server aggregates client updates based on the magnitude of the drift to ensure global consistency.

**Compliance With Llm Reviewing Policy:**

Affirmed.

**Final Justification:**

The rebuttal has addressed all my concerns, and I am willing to increase the score.

**Key Questions For Authors:**

Regarding Eq. (5), the Sinkhorn distance relies on feature representations from the local model $w_k^t$. Since parameters evolve during training, could the authors clarify if drift detection is performed just once using the initial frozen model, or dynamically? If dynamic, how does the method account for shifting feature spaces?

**Limitations:**

Yes.

**Strengths And Weaknesses:**

Strengths：

1. The paper is clearly structured, with a natural flow from problem setup to method and experiments, which makes it straightforward to follow the overall idea.
2. The paper studies asynchronous concept drift in FCL, a well-motivated and realistic scenario where clients evolve differently over time and are not adequately addressed by existing methods.
3. The method RC-FCL is aligned with the problem, with its modules designed to address the challenges of asynchronous concept drift by improving consistency across clients under evolving data distributions.

Weaknesses：

1. Since Federated Learning often targets resource-constrained edge devices, calculating optimal transport distances and training a local CGAN  might introduce latency.

2. The readability could be further enhanced by clarifying some mathematical notations that are currently slightly ambiguous. For example, regarding Eq. (6), it would be helpful to explicitly state whether the discriminator $D_k$ represents a frozen snapshot from the previous task or a model dynamically updated during the current step.

---

> ### Author Rebuttal · Authors · 2026-03-30
>
> > **W1.Concern about resource constraint**
>
> **R1:**  Thank you for raising this concern. We conducted an additional overhead experiment.
>
> |Method|Peak Memory (MB)|Time (s)/round|Upload/round (MB)|Download/round (MB)|
> |-|-:|-:|-:|-:|
> |ReFed|247.56|325|44.72|44.72|
> |GLFC|312.91|367|44.72|44.72|
> |CAN[1]|301.36|405|56.02|56.02|
> |RC-FCL|295.22|389|56.11|56.11|
>
> The generative module is a multipurpose choice, as it is used for replay and drift adaptation, while discriminator is reused. Compared with memory buffer methods, we do not require storing raw data. Although we introduce a slight additional cost, it remains a cost-effective design.
>
> > **W2. Issues about mathematical notations and ambiguous**
>
> **R2:** We appreciate this detailed feedback for improving clarity. In Eq. (6), the discriminator $D_k^t$  is intended to denote the discriminator learned from the previous task $t$, which serves as the reference when weighting samples for task $t+1$, rather than a discriminator being dynamically updated during the weighting step itself. Specifically, during task $t$, each client trains a local CGAN, yielding  $G_k^t$ and $D_k^t$ on the current data distribution alongside the classifier. When task $t+1$ arrives, $D_k^t$ is used to assess the novelty of incoming samples: a lower output score indicates that a sample deviates more from the distribution learned at task $t$, and thus receives a higher weight in Eq. (6). The updated discriminator for the current task is obtained only in the subsequent CGAN training phase. This design ensures that sample weighting is based on a stable pre-drift reference. We will clarify the notation in the revision.
>
> > **Q1.Concern about feature representations**
>
> **A1:** Thank you for this insightful question. In RC-FCL, drift detection is not intended to be performed continuously while the local classifier is being updated within the same optimization step. Rather, at a given communication round (or the arrival of a new task), each client first receives the current model snapshot $w_k^t$ from the server and uses this snapshot to extract feature representations for both the current local data and the synthetic reference data when computing Eq. (5). The subsequent local model update is performed only after drift detection and calibration weights have been determined.
>
> Therefore, the method is dynamic across rounds, but the feature space used in Eq. (5) is fixed within each detection step. This avoids ambiguity caused by a moving representation space during distance computation. Importantly, both the current local samples and the synthetic pre-drift references are embedded by the same client model snapshot $w_k^t$, so the Sinkhorn distance is always measured in a shared feature space at that step. As the model evolves in later rounds, drift detection is recomputed using the updated snapshot, which allows the reference comparison to adapt over time. We will clarify this process more explicitly in the revision.
>
> > **References**
>
> [1]CAN: Leveraging Clients As Navigators for Generative Replay in Federated Continual Learning. (ICML 2025)

---

> > ### Author Rebuttal · Reviewer_uYMd · 2026-04-01
> >
> > Thank you for the author's reply. It has basically solved all my concerns. I am willing to increase my score.

---

> > > ### Author Response · Authors · 2026-04-04
> > >
> > > Thank you very much for your helpful comments. We will continue refining the manuscript accordingly.

---

### Official Review · Reviewer_Scv9 · 2026-03-12

**Soundness:** 3
**Presentation:** 2
**Significance:** 2
**Originality:** 2
**Overall Recommendation:** 4
**Confidence:** 4

**Summary:**

The paper presents RC-FCL, which addresses asynchronous concept drift in federated continual learning. Authors propose a retrospective calibration framework built around a shared conditional GAN, a drift detection module that compares local feature distributions against synthetic references, and a drift-aware aggregation strategy that up-weights clients experiencing larger distribution shifts.

**Compliance With Llm Reviewing Policy:**

Affirmed.

**Final Justification:**

The authors provided a satisfactory response to my points. While a more robust evaluation on more diverse domains would be appreciated, as well as an assessment of real-world concept drift scenarios in addition to synthetic datasets, I have changed my recommendation to weak accept.

**Key Questions For Authors:**

- The baseline selection omits several 2024-2025 FCL and concept drift adaptation methods that are cited in the related work but excluded from experiments. What is the justification for this omission, and how does RC-FCL perform against these more recent methods?
- How does RC-FCL perform on non-visual or more realistic benchmarks?
- How does RC-FCL perform when concept drift is gradual rather than abrupt?
- What is the computational cost of CGAN training per client per round?

**Limitations:**

Yes

**Strengths And Weaknesses:**

**Strengths**
- Sinkhorn distance is shown to be an advantageous choice for distribution comparison, and authors compare it against MMD, KL divergence and cosine similarity to validate their choice.
- Writing is generally clear and algorithms are provided to aid reproducibility.
- Ablation studies are appreciated and isolate the contribution of drift detection vs adaptation.

**Weaknesses**
- Compared baselines are not SOTA, and the majority of comparisons are against methods published in 2021-2023. Re-Fed (2024) is the most recent baseline. The authors cite several 2024-2025 works in the related work section, but do not include them as experimental baselines.
- Selected benchmarks (CIFAR-10, CIFAR-100, Tiny-ImageNet, Digit10) are weak, limited to small-scale visual classification tasks. There are no non-vision datasets, time-series benchmarks, or domain-shift benchmarks that might have strengthened the generalizability.
- Experimental results are very low (e.g. 26.69% on Tiny-ImageNet), and do not reflect the results reported in other cited FCL papers. The authors should explain why this is the case.
- The CGAN introduces substantial system overhead. Each client trains both a generator and a discriminator in addition to the classifier. Table 2 reports communication rounds, but there is no analysis of computational cost per round, wall-clock training time, or memory footprint, all of which are critical for real-world deployment.
- RC-FCL introduces three hyperparameters that require careful tuning. Although Figure 6 provides sensitivity curves, the default values are reported without guidance on how to select them in practice when ground truth drift labels are unavailable.

---

> ### Author Rebuttal · Authors · 2026-03-30
>
> Thank you for these valuable comments.
>
> > **W1 & Q1.Baseline.**
>
> **R1:** We follow widely adopted FCL baselines, which remain standard and meaningful for comparison, as also reflected in several recent works from 2025 [1–4] that include methods such as FedEWC, FedLwF, TARGET, GLFC, ReFed, and FedWeIT. We also add recent 2025 baselines：
>
> | Dataset|CIFAR-100|||Tiny-ImageNet|||
> |-|-:|-:|-:|-:|-:|-:|
> |Partition|IID|0.5|0.1|IID|0.5|0.1|
> |Metric|Acc(↑)/FS(↓)| Acc(↑)/FS(↓)|Acc(↑)/FS(↓)|Acc(↑)/FS(↓)|Acc(↑)/FS(↓)|Acc(↑)/FS(↓)|
> |CAN[4]|28.72/39.67|25.67/38.82|24.93/40.78|24.29/37.91|22.18/39.96|21.07/40.46|
> |FedTA[3]|32.14/34.92|28.68/37.10|24.54/39.03|25.06/40.12|21.71/41.25|20.92/43.19|
> |FedCBDR[2]|26.97/38.76|23.72/39.88|20.54/43.25|21.68/41.05|19.76/43.37|19.53/42.31|
> |RC-FCL|35.23/29.65|32.01/35.24|29.77/37.59|26.69/39.42|24.23/38.49|22.62/40.11|
>
> > **W2 & Q2. Benchmark.**
>
> **R2:** We choose four public datasets, which have been widely adopted in existing studies on FCL and concept drift under FL. Recent works [2], [3], and [4] all use CIFAR-100, and some of them also include Tiny-ImageNet and CIFAR-10. We do not use ImageNet-R because it has 200 classes, the same as Tiny-ImageNet, but a smaller data scale. Earlier works [4] [5] also use CIFAR-10/100. We follow the settings used in most prior work, which enables a fair comparison with existing methods.
>
> Most existing FCL methods are still mainly evaluated on vision datasets. Therefore, in this work, we do not extend the evaluation to non-vision modalities such as text or time series. However, concept drift in other modalities, especially time series, will be an important direction for our future work.
>
> > **W3. Experimental results**
>
> **R3:** Our setting is significantly more challenging than standard FCL. Existing methods, such as TARGET[5], ReFed[6], CAN[4], and FedCBDR[2] are evaluated mainly under standard incremental streams, where the key challenge is forgetting caused by newly arriving tasks or non-IID data. For example, TARGET reports 36.31 on CIFAR-100 with 5 tasks; Re-Fed reports about 25.61–29.90 on CIFAR-100 with 10 tasks; CAN reports 34.41(IID) on CIFAR100 with 10 tasks. FedCBDR reports 45.11 and 46.51 on CIFAR-100 with 10 tasks under different heterogeneity levels.
>
> We introduce asynchronous concept drift into FCL, which makes the experimental setting considerably more challenging, as the model must adapt to learned classes that drift asynchronously across clients and over time. As a result, all baselines show performance drops in our setting, which in turn suggests that asynchronous concept drift is indeed present in FCL and constitutes an important problem that deserves further study.
>
> > **W4 & Q4.Computational cost**
>
> **R4:** We report overhead below, run on an RTX 4060 GPU.
>
> |Method|Peak Memory (MB)|Time (s)/round|Upload/round (MB)|Download/round (MB)|
> |-|-:|-:|-:|-:|
> |ReFed|247.56|325|44.72|44.72|
> |GLFC|312.91|367|44.72|44.72|
> |CAN[4]|301.36|405|56.02|56.02|
> |RC-FCL|295.22|389|56.11|56.11|
>
> The generative module is a multipurpose choice, as it is used for replay and drift adaptation, while discriminator is reused. Compared with memory buffer methods, we do not require storing raw data. Although we introduce a slight additional cost, it remains a cost-effective design.
>
> > **W5. Hyperparameters selection**
>
> **R5**: As shown in Fig. 6 and Hyperparameters analysis, setting the threshold to 0.4 yields the best performance on both CIFAR100 and Tiny-ImageNet, while 0.3 also achieves competitive results. Our method remains consistently superior across this range, indicating that its effectiveness is not substantially affected by the parameter choice; therefore, 0.3–0.4 can be regarded as a reasonable and robust range in practice, with further tuning likely offering additional gains.
>
> **Q3**: Drift Type
>
> **A3:** Two common ways to simulate concept drift: transformation and label swapping. We adopt the former because it better reflects realistic drift in visual data, where image changes are usually gradual and continuously accumulated. By contrast, label swapping introduces a much more abrupt shift, where one concept is directly replaced by another at the semantic level. This type of change is less consistent with how concept drift typically occurs in real-world image scenarios. We will include results under the label-swapping setting in the revision.
>
> > **References**
>
> [1]pFedMxF: Personalized Federated Class-incremental Learning with Mixture of Frequency Aggregation. (CVPR 2025)
>
> [2]Class-wise Balancing Data Replay for Federated Class-Incremental Learning. (NeurIPS 2025)
>
> [3]Handling Spatial-Temporal Data Heterogeneity for Federated Continual Learning via Tail Anchor. (CVPR 2025)
>
> [4]CAN: Leveraging Clients As Navigators for Generative Replay in Federated Continual Learning. (ICML 2025)
>
> [5]TARGET: Federated Class-Continual Learning via Exemplar-Free Distillation. (ICCV 2023)
>
> [6]Towards Efficient Replay in Federated Incremental Learning. (CVPR 2024)

---

> > ### Author Rebuttal · Reviewer_Scv9 · 2026-04-01
> >
> > Thank you to the authors for providing a thorough response. It has resolved most of my concerns. However, on the benchmark point, I maintain that additional domains are valuable for extending the approach's generalisability. While I understand that the image datasets presented are historically common in continual learning, it is becoming more regular to include datasets that show applicability to other domains, e.g. [1], [2]. This helps to further a research perspective of continual learning beyond image classification tasks.
> >
> > [1] Zhong, Zhengyi, et al. "Sacfl: Self-adaptive federated continual learning for resource-constrained end devices." IEEE Transactions on Neural Networks and Learning Systems (2025).
> >
> > [2] Luopan, Yaxin, et al. "Loci: Federated continual learning of heterogeneous tasks at edge." IEEE Transactions on Parallel and Distributed Systems 36.4 (2025): 775-790.

---

> > > ### Author Response · Authors · 2026-04-03
> > >
> > > Thank you very much for your suggestions regarding benchmark diversity. We fully understand your concerns and conduct experiments for text classification tasks.
> > >
> > > >**Dataset**
> > >
> > > Based on the benchmarks in the references provided, we selected the THUCNews[1] dataset used in SacFL [2]. THUCNews contains 740,000 news documents collected from the Sina News RSS subscription channels between 2005 and 2011. These news articles are categorized into 14 classes: finance, lottery, real estate, stocks, home furnishing, education, technology, society, fashion, current politics, sports, constellation, games, and entertainment. We randomly sample 5,000 instances from each of 10 selected classes for continual learning, while the remaining 4 classes (constellation, real estate, finance, current politics) are used for drift.
> > >
> > > >**Text concept drift simulation**
> > >
> > > Following [3] [4], we simulate concept drift using two settings:(1)Class swap.For a class in previously learned tasks, once concept drift occurs, its samples are replaced by samples from any one of the remaining four unused classes, while the original label is kept unchanged.(2)Class shift, involves integrating different types of content to bring about a shift in the distribution within a class. This setting reflects real-world topic evolution, for example, when financial news gradually blends stock market reports with broader economic commentary. In our experiments, once concept drift occurs in a class, we merge its text content with that from one of the remaining four unused classes; for instance, real estate samples are concatenated with home furnishing samples.
> > >
> > > >**Configuration**
> > >
> > > We adopt 5-tasks setting, Dirichlet distribution with 0.5/0.1  to simulate data heterogeneity, threshold $\tau$ = 0.4, $\epsilon$ =0.05, $K$ = 20 clients with an active ratio 0.4, batch size 32, learning rate 0.001. For efficiency, we use 10 local training epochs, and 50 communication rounds per task.
> > >
> > > Following the references [2] [5] you provided, we use TextCNN as classifier, and its lightweight architecture is suitable for federated learning. Due to the discrete of text, we implement the conditional generator in the continuous feature space produced by the TextCNN encoder, rather than in the raw text space. This modification preserves the mechanisms of RC-FCL: the generator synthesizes class-conditional pre-drift references for previously learned tasks, and the current local distribution is compared against these references in feature space for drift detection. Specifically, the generated latent features are used for Sinkhorn drift detection and replay.  The discriminator is reused as an importance estimator in feature space for local calibration. At the server side, the uploaded client updates are still combined using the original drift-aware aggregation strategy, where aggregation weights are determined according to the detected drift statistics. In this way, the text extension preserves the original functional structure of RC-FCL, including conditional reference generation, feature space drift detection, discriminator local calibration, and drift-aware global aggregation.
> > >
> > > >**Results**
> > >
> > > The experimental results are shown in the table below. Our method achieves competitive performance under both simulated concept drift settings.
> > >
> > > ||Class swap||Class shift ||
> > > |-|-:|-:|-:|-:|
> > > | Partition |0.5|0.1| 0.5|0.1|
> > > | Metric| Acc(↑)/FS(↓) | Acc(↑)/FS(↓) | Acc(↑)/FS(↓) | Acc(↑)/FS(↓) |
> > > | SacFL[2]|49.91/29.05|45.72/31.12|52.18/27.46|49.66/33.15|
> > > | ReFed[6]|47.67/35.85|44.83/34.92|50.76/29.38|50.31/30.96|
> > > | CAN[7]|50.50/29.78 |50.12/30.15|49.09/31.77|47.28/35.31|
> > > | RC-FCL|56.05/26.49|53.29/28.32|54.25/25.01|52.36/30.43|
> > >
> > > Due to time constraints, we are unable to compare our method with more baselines. We will conduct experiments on the remaining benchmarks listed in the references provided and other domain benchmarks, and will include the results in the revised version. We sincerely appreciate your valuable suggestions, which have helped us improve this work.
> > >
> > > >**References**
> > >
> > > [1] A comparison and semi-quantitative analysis of words and character-bigrams as features in Chinese text categorization. (ACL 2006)
> > >
> > > [2] SacFL: Self-adaptive federated continual learning for resource-constrained end devices.(IEEE Transactions on Neural Networks and Learning Systems 2025)
> > >
> > > [3] Concept drift adaptation in text stream mining settings: A systematic review
> > > (ACM Transactions on Intelligent Systems and Technology 2025)
> > >
> > > [4] Unsupervised Concept Drift Detection from Deep Learning Representations in Real-time
> > > (IEEE Transactions on Knowledge and Data Engineering 2025)
> > >
> > > [5] Loci: Federated continual learning of heterogeneous tasks at edge
> > > (IEEE Transactions on Parallel and Distributed Systems 2025
> > >
> > > [6] Towards Efficient Replay in Federated Incremental Learning. (CVPR 2024)
> > >
> > > [7] CAN: Leveraging Clients As Navigators for Generative Replay in Federated Continual Learning. (ICML 2025)

---

### Official Review · Reviewer_W7v5 · 2026-03-13

**Soundness:** 2
**Presentation:** 3
**Significance:** 3
**Originality:** 2
**Overall Recommendation:** 3
**Confidence:** 4

**Summary:**

This paper aims to solve the asynchronous concept drift problem in Federated Continual Learning (FCL), where different clients experience distribution shifts at different times and magnitudes. To address this issue, the paper proposes RC-FCL, a retrospective calibration framework that integrates drift detection and adaptive learning in FCL. Specifically, a shared conditional generative model synthesizes class-conditional reference distributions of previously learned data. Clients compare local features with synthetic references using Sinkhorn distance to detect class-level drift. Then, drifted samples are assigned adaptive weights using a discriminator trained on pre-drift distributions. Finally, the server aggregates client updates using weights determined by drift magnitude and prevalence across clients. Experiments on CIFAR-10/100, Tiny-ImageNet, and Digit10 demonstrate improvements over several federated continual learning baselines in terms of accuracy and forgetting score.

**Compliance With Llm Reviewing Policy:**

Affirmed.

**Final Justification:**

After carefully reading both the paper and the authors’ rebuttal, I still believe that ICML requires a more principled theoretical foundation to justify the proposed design. Therefore, I will maintain my original score, while sincerely appreciating the authors’ response.

**Key Questions For Authors:**

1. The experiments mainly simulate drift using image transformations. Can the framework handle semantic drift or label distribution drift (e.g., class redefinition or appearance of new concepts)?
2. Is there any theoretical analysis supporting the stability or convergence of the drift-aware aggregation strategy?
3. The drift detection relies on a threshold parameter $\tau$. How sensitive is performance to this hyperparameter, and how should it be selected in practice?

**Limitations:**

Please talk about the limitations of this work

**Strengths And Weaknesses:**

Strengths:
1. The paper clearly identifies asynchronous concept drift as a practical challenge in FCL, and I also believe this is an important and realistic problem that deserves further study.
2. Using a conditional generator to synthesize historical distributions is an interesting design because clients cannot store or share past data in many FL scenarios. This mechanism allows drift detection without violating privacy constraints.
3. The paper is well-organised and well-written

Weaknesses:
1. The proposed method integrates several existing techniques, such as generative replay with GANs, Sinkhorn distance for distribution comparison, weighted aggregation, and discriminator-based sample weighting. On the one hand, this makes the core algorithmic novelty appear incremental. On the other hand, the framework lacks a unified theoretical analysis or principled constraint to justify how these components interact or contribute to the overall learning objective.
2. The method relies heavily on the quality of synthetic reference distributions generated by CGAN. If the generator fails to approximate the pre-drift distribution accurately, drift detection accuracy could degrade. This issue is not sufficiently analyzed.
3. In experiments, concept drift is simulated using image transformations such as noise, rotation, and erasing. While controlled, these transformations mainly represent covariate shifts, and may not fully capture more realistic semantic or task-level concept drift.
4. Training and maintaining both generator and discriminator models on clients introduces additional computation and communication overhead. A deeper analysis of memory consumption, training time and generator stability would improve evaluation.
5. The baseline methods used for comparison are mostly from work published before 2024. The authors are encouraged to include comparisons with more recent work (e.g., 2025 studies). In addition, the problem of asynchronous drift has been widely discussed in the data stream learning literature, such as [1-2], it would be beneficial for the authors to discuss the relationship and differences between the proposed setting and existing asynchronous drift formulations in the data streaming literature. Such a discussion would help readers better understand the positioning and novelty of the proposed approach.
[1] Drift-aware collaborative assistance mixture of experts for heterogeneous multistream learning. arXiv preprint arXiv:2508.01598 (2025). [2] Multistream regression with asynchronous concept drift detection. 2017 IEEE International Conference on Big Data.
6. In terms of reproducibility, it is recommended that the authors release the dataset preprocessing details and the code implementation to ensure the reproducibility of the experimental results.

---

> ### Author Rebuttal · Authors · 2026-03-30
>
> Thank you for your thoughtful comments.
>
> > **W1. & Q2 Novelty and theory**
>
> **R1:** We propose asynchronous concept drift in FCL, where different clients experience concept drift on different classes at different times, making drift sparse and easily confounded with client heterogeneity. Privacy constraints also preclude storing raw historical data as pre-drift references.
>
> We propose a compact and resource-efficient retrospective calibration framework. CGAN (rather than GAN) is used for replay and generating the reference for specific learned classes for detection. Discriminator is reused as a drift-aware importance estimator, and the server performs drift-aware aggregation, so distribution changes caused by concept drift are not obscured by heterogeneity during aggregation.
>
> A rigorous analysis is challenging because local objectives vary across both clients and time under asynchronous drift, violating standard assumptions used in convergence proofs, such as synchronized updates and shared drift patterns. Consistent with recent FCL works[1][2], we therefore focus on empirical validation and leave formal analysis to future work
>
> > **W2. Synthetic reference**
>
> **R2:** We use CGAN as a simple and widely validated generator to demonstrate the effectiveness of conditional generation for both drift detection and replay. CGAN is not the only option, it can be replaced, such as [3],  for better reference.
>
> > **W3 & Q1. Drift simulation and applicability**
>
> **R3:** Two main methods for simulating concept drift: [4] applies **transformations** to get images that have a different style but share the same labels；[5] applies **label swapping**. We adopt the former because, in real-world scenarios, concept drift usually occurs gradually, while full label replacement is rare. We will add label swapping results in the discussion section.
>
> > **W4. Overhead**
>
> **R4:** We report overhead below, with all experiments run on an RTX 4060 GPU.
>
> |Method|Peak Memory (MB)|Time (s)/round|Upload/round (MB)|Download/round (MB)|
> |-|-:|-:|-:|-:|
> |ReFed|247.56|325|44.72|44.72|
> |GLFC|312.91|367|44.72|44.72|
> |CAN[6]|301.36|405|56.02|56.02|
> |RC-FCL|295.22|389|56.11|56.11|
>
> The generative module is a multipurpose choice, as it is used for replay and drift adaptation, while discriminator is reused. Compared with memory buffer methods, we do not require storing raw data. Although we introduce a slight additional cost, it remains a cost-effective design.
>
> > **W5. Baseline and work positioning**
>
> **R5:** We add relevant baselines published in 2025：
>
> | Dataset|CIFAR-100|||Tiny-ImageNet|||
> |-|-:|-:|-:|-:|-:|-:|
> |Partition|IID|0.5|0.1|IID|0.5|0.1|
> |Metric|Acc(↑)/FS(↓)| Acc(↑)/FS(↓)|Acc(↑)/FS(↓)|Acc(↑)/FS(↓)|Acc(↑)/FS(↓)|Acc(↑)/FS(↓)|
> |CAN[6]|28.72/39.67|25.67/38.82|24.93/40.78|24.29/37.91|22.18/39.96|21.07/40.46|
> |FedTA[2]|32.14/34.92|28.68/37.10|24.54/39.03|25.06/40.12|21.71/41.25|20.92/43.19|
> |FedCBDR[1]|26.97/38.76|23.72/39.88|20.54/43.25|21.68/41.05|19.76/43.37|19.53/42.31|
> |RC-FCL|35.23/29.65|32.01/35.24|29.77/37.59|26.69/39.42|24.23/38.49|22.62/40.11|
>
> Asynchronous concept drift and multistream learning both address dynamic distribution evolution across heterogeneous sources. Our setting operates under privacy constraints where data sharing is prohibited and global awareness is unavailable, necessitating data-free drift detection and adaptation rather than direct cross-source information sharing.
>
> > **W6. Reproducibility**
>
> We promise to release the code if the work is accepted.
>
> > **Q3. Threshold**
>
> **A3:** The table reports accuracy for $\tau$ at 0.1, 0.4 and 0.9; the full sweep from 0.1 to 0.9 is provided in Appendix C.1. Performance is relatively stable for τ around 0.4 on both datasets. This suggests RC-FCL is not highly sensitive to τ within a moderate range.”
>
> |$\tau$|CIFAR100|||Tiny-ImageNet|||
> |-|-:|-:|-:|-:|-:|-:|
> |Partition|IID|0.5|0.1|IID| 0.5| 0.1|
> |0.1|22.35|19.67|18.93|15.17|14.28|11.73|
> |0.4|35.23|32.01|29.77|26.69|23.57|22.62|
> |0.9|19.21|18.57|18.26|12.97|11.33|10.38|
>
> > **Llimitations**
>
> This study mainly follows the most widely used setting in FCL and is validated on vision datasets. This limits the generalizability of the method to more complex scenarios. Future work will explore evaluations on larger-scale and more diverse datasets to further extend the applicability.
>
> > **References**
>
> [1]Class-wise Balancing Data Replay for Federated Class-Incremental Learning. (NeurIPS 2025)
>
> [2]Handling Spatial-Temporal Data Heterogeneity for Federated Continual Learning via Tail Anchor. (CVPR 2025)
>
> [3]Distilling Diffusion Models into Conditional GANs. (ECCV 2024)
>
> [4]FedNN: Federated learning on concept drift data using weight and adaptive group normalizations. (PR 2024)
>
> [5]Flash: Concept Drift Adaptation in Federated Learning. (ICML 2023)
>
> [6]CAN: Leveraging Clients As Navigators for Generative Replay in Federated Continual Learning. (ICML 2025)

---

> > ### Author Rebuttal · Reviewer_W7v5 · 2026-04-02
> >
> > I appreciate the authors’ effort in clarifying the motivation of the setting, adding newer baselines, and providing additional overhead numbers. After carefully reading both the paper and the response, some of my concerns are still unresolved:
> >
> > 1. On the central issue of novelty and theoretical grounding, my concern is not simply whether the overall pipeline is reasonable, but whether the paper provides a principled formulation that justifies why the combination of its components should work together. In other words, this requires some level of theoretical analysis or justification. I do not question the practical reasonableness of the proposed design or the empirical performance; however, for a venue like ICML, I believe a more insightful theoretical foundation is expected. The rebuttal does not fully address this concern. The authors mainly restate the design rationale and argue that rigorous analysis is difficult, and therefore focus on empirical validation while leaving theoretical analysis to future work.
> >
> > 2. Regarding Response to W2, the rebuttal answers this mainly by stating that CGAN is “simple and widely validated” and could in principle be replaced by stronger generators. That does not address the actual concern, which is whether the current method is robust when the reference distribution is imperfect, miscalibrated, or drifts away from the true pre-drift distribution. Since the detection score in Eq. (5) directly depends on these synthetic references, this dependency should be analyzed rather than deferred by saying another generator could be used. This is not merely an implementation detail, but is central to the validity of the proposed drift detection mechanism.

---

> > > ### Author Response · Authors · 2026-04-04
> > >
> > > We thank the reviewer for the continued engagement. We now directly address the concern of RC-FCL’s component synergy.
> > >
> > > > **Problem Formulation: ideal drift  calibration and its unobservability**
> > >
> > > In standard FCL, class distributions remain stable. Asynchronous drift disrupts this, creating severe heterogeneity across clients, classes, and drift magnitudes. Consequently, different clients optimize conflicting target distributions, destroying global convergence.
> > >
> > > We decompose the local loss of client $k$ at class-level granularity and formulate the local training objective in Eq.(2).
> > > Eq.(2) introduces $\gamma_k^{t}(y) \in [0, 1]$, and the ideal drift-calibration strength noted as $\hat{\gamma}_k^t(y)$. $\gamma=0$ means replay is sufficient, $\gamma=1$ means complete shift, and intermediate values require fine-grained calibration. However, strictly under FCL privacy constraints, the ideal $\hat{\gamma}$ is unobservable since clients cannot access the global historical distribution $P^{t-1}$.
> > >
> > > > **RC-FCL Design**
> > >
> > > RC-FCL is governed by a principle: approximating the ideal $\hat{\gamma}_k^t(y)$ under FCL, while systematically suppressing the inevitable estimation errors at both local and global levels.
> > >
> > > > **CGAN and Sinkhorn**
> > >
> > > The CGAN and Sinkhorn distance serve as functional components that work in concert under privacy constraints to compute a class-level approximation $\gamma_k^t(y)$ (Eq.5, drift score), but inevitably introduce errors due to generation quality and entropic regularization.
> > >
> > > > **Discriminator Weighting**
> > >
> > > To prevent class-level detection errors from directly corrupting local optimization, we employ the discriminator to bridge the gap between class-level and instance-level granularity, providing a second-pass filter at the instance level. The weighting mechanism (Eq. 6) provides a risk control: If $\gamma$ overestimates drift (False Positive), pre-drift samples still yield high $D_k(x_i)$, driving weight $\alpha_i \to 0$ and automatically dampening erroneous gradients. Underestimations merely delay adaptation without introducing destructive gradients.
> > >
> > > > **Prevalence-aware aggregation**
> > >
> > > While the local discriminator provides an instance-level buffer, its capability is inherently bounded by data heterogeneity and the quality of generated data, which may cause local filters to fail, resulting in isolated detection noise across clients. Prevalence-aware aggregation serves statistically as a majority-vote consistency check, providing global-level correction that does not depend on any single client's generation quality, ensuring that degradation at individual clients does not propagate to the global model.
> > >
> > > We adopt a relatively conservative design: selective amplification, rather than selective suppression—it never discards any client's information, but instead determines the degree of amplification based on the cross-client consistency of drift. The rapid scaling based on prevalence ensures that even in extreme cases, real drift is not completely overwhelmed.
> > >
> > > The various modules work in concert to approximate the ideal drift calibration and perform step-by-step error correction, ensuring that the model adapts to asynchronous drift under FCL without compromising stability.
> > >
> > > > **Generation quality experiments**
> > >
> > > We demonstrate how generation quality bounds our performance on CIFAR-100. To isolate this variable without disrupting the baseline replay of non-drifted classes (which reducing CGAN epochs would cause), we intentionally corrupted the generated distributions using spatial transformations distinct from the drift simulations in our submission (10-30%, 30-60%, >60% area).
> > >
> > > |Dataset|CIFAR-100| ||
> > > | - | -: | --: | -- |
> > > | Area | <30% | [30%,60%] |>60%|
> > > | Metric |   Acc(↑) |   Acc(↑) |Acc(↑) |
> > > | CAN[1] | 15.67 | 15.03 |14.70 |
> > > | FedCBDR[2] | 17.46 | 14.61 |13.64 |
> > > | ReFed[3] | 17.58 | 17.27 |15.38 |
> > > | RC-FCL | 29.54 | 24.89 |14.25 |
> > >
> > > RC-FCL maintains a substantial advantage under minor-to-moderate degradation. Only under severe degradation (>60%) does the error exceed the dampening capacity, degenerating to baselines.
> > >
> > > > **Label swapping experiments**
> > >
> > > As promised in R3, we present here the experimental results of label swapping. We conducted these experiments on the CIFAR-100 dataset with Dirichlet distributions of 0.5 and 0.1.
> > >
> > > |Dataset|CIFAR-100 | |
> > > | - | -: | -: |
> > > | Partition| 0.5 | 0.1 |
> > > | Metric| Acc(↑)/FS(↓) |  Acc(↑)/FS(↓) |
> > > | CAN[1] | 23.42/40.45 | 23.04/39.52 |
> > > | FedCBDR[2] |  25.63/38.21 | 22.55/41.06 |
> > > | ReFed[3] |27.93/36.17 | 25.67/38.83 |
> > > | RC-FCL| 34.96/33.78 | 33.16/37.20 |
> > >
> > > We thank the reviewer again for the insightful comments and will clarify these in the revision.
> > >
> > > > **References**
> > >
> > > [1]CAN: Leveraging Clients As Navigators for Generative Replay in Federated Continual Learning. (ICML 2025)
> > >
> > > [2]Class-wise Balancing Data Replay for Federated Class-Incremental Learning. (NeurIPS 2025)
> > >
> > > [3]Towards Efficient Replay in Federated Incremental Learning. (CVPR 2024)

---

### Decision · Program_Chairs · 2026-04-30

**Decision:**

Accept (regular)

**Comment:**

This paper propose a retrospective calibration framework that effectively addresses the practical challenge of asynchronous concept drift in Federated Continual Learning through generative replay and adaptive weighting, named RC-FCL. Reviewers appreciate the clear structure of the manuscript, the well-motivated problem setting, and the strong empirical results demonstrated across diverse datasets and drift scenarios. Although initial concerns were raised regarding the computational overhead of the local generative models and the reliance on somewhat dated baselines, the authors provided clarifications and additional analyses during the rebuttal. However, there are still pressing concerns by the reviewer W7v5 for lack of principled theoretical formulation. Overall, considering the thorough experimental validation and the successful resolution of reviewer questions, the AC recommends to accept the paper to ICML 2026.